# OpenReview forum: "White-Box Auditing of Large Language Model Unlearning"
_ICLR.cc/2026/Conference — Submitted to ICLR 2026_

### Official Review · Reviewer_oink · 2025-10-17

**Soundness:** 1
**Presentation:** 2
**Contribution:** 1
**Rating:** 2
**Confidence:** 4

**Summary:**

The paper addresses privacy concerns in large language models (LLMs) by proposing a white-box auditing framework to evaluate whether unlearning methods truly erase sensitive personal information (PI) or merely conceal it. The authors create a synthetic dataset called Fake Personal Information (FPI), containing attributes like year of birth, blood type, postcode, and social insurance number. They further introduce a restricted inverse greedy (RIG) decoding strategy to recover supposedly forgotten PI from the model's logits, in order to audit the unlearning effectiveness.

**Strengths:**

1. The structure of the paper is clear.
2. It is an important task to correctly audit the current unlearning method.
3. The PI inference task is suitable for unlearning auditing, considering PI are easily verifiable.

**Weaknesses:**

1. The contributions are limited. The proposed dataset is similar to previous TOFU or SynthPAI. Besides, traditional datasets for PI inference could be easily modified (or even directly used) to unlearning auditing tasks.
2. The paper is a bit over-claimed. Considering the auditing method may be limited to PI datasets (or similar tasks with verifiable answers) and fine-tuning unlearning methods, it is not a unified auditing method.
3. The proposed method is lack of empirical score to comprehensively quantify the ability for a given unlearning method. Besides, the idea itself lacks stronger (even theoretical guarantee), and it is intuitive for rigorous auditing task.
4. The key finding that existing unlearning methods can not robustly erase the target knowledge is not a new conclusion, many recent works have reveal this point. However, the authors fail to fully discuss and compare with them.

**Questions:**

1. What are the differences or novelty of the proposed dataset, compared with previous works or tradition PI datasets?
2. Can the auditing method be applied to broader tasks (e.g., open-ended generation), or other types unlearning methods?
3. The proposed auditing method needs stronger guarantee.
4. What are new findings compared to previous works, e.g., Inexact Unlearning Needs More Careful Evaluations to Avoid a False Sense of Privacy?

**Details Of Ethics Concerns:**

No.

---

> ### Author Response · Authors · 2025-11-28
>
> >- The contributions are limited. The proposed dataset is similar to previous TOFU or SynthPAI. Besides, traditional datasets for PI inference could be easily modified (or even directly used) to unlearning auditing tasks.
> >- What are the differences or novelty of the proposed dataset, compared with previous works or tradition PI datasets?
>
>
> We thank the reviewer for raising this question regarding the novelty of the **FPI** dataset. While our synthetic generation pipeline is inspired by TOFU and appears to have some similarity with SynthPAI, **FPI is designed for a fundamentally different task** and supports evaluations that existing datasets cannot provide.
>
> **Relation to TOFU (Knowledge Unlearning)**
>
> The resemblance between FPI and TOFU is limited to the _synthetic dataset generation procedure_. The _research objectives_ differ substantially:
> - TOFU targets _knowledge unlearning_: removing factual or conceptual knowledge. The data for knowledge unlearning are usually QA pairs with _open-ended answers_, because a single piece of knowledge may be expressed in many valid ways. As a result, no single ground-truth answer exists, and evaluation relies on text-similarity metrics such as ROUGE.
> - FPI, in contrast, targets _personal-information unlearning_, where attributes (e.g., SIN, phone numbers, addresses) are **structured and uniquely defined**. This allows **precise, attribute-specific metrics** to quantify what PI remains in a model’s outputs.
>
> **PI unlearning and knowledge unlearning differ in data properties**
>
> - Personal information for different individuals—or different PI of the same individual—are **not entangled**. One cannot infer person A’s SIN from person B’s SIN, nor infer an SIN from a birthday.
> - Factual knowledge, by contrast, is inherently **entangled**: removing one fact often affects related knowledge in the retain set, making forget/retain partitioning ambiguous. For example, knowledge of how to conduct a cyberattack is deeply intertwined with general cybersecurity concepts, and seemingly benign pieces of information, when combined, may enable the reconstruction of harmful knowledge, as shown in the paper _The WMDP Benchmark: Measuring and Reducing
> Malicious Use With Unlearning_.
>
> Due to these differences,  **knowledge-unlearning datasets like TOFU cannot be used to study PI unlearning**, and results from knowledge unlearning do not transfer. This motivates constructing FPI as a dataset with:
> - Multiple heterogeneous PI attributes
> - Attribute-specific evaluation metrics
> - A clean and controllable forget–retain partition
> To our knowledge, no existing dataset—including TOFU—supports **systematic auditing of PI unlearning** across diverse attribute types.
>
> **Comparison with SynthPAI**
>
> Although SynthPAI also includes “personal information,” the underlying task is distinct.
> - **SynthPAI** is designed for _author profiling_: inferring demographic or personal traits of an **unknown individual** from their writing. The PI labels are _not explicitly present_ in the input text.
> - **FPI**, by contrast, is designed for _auditing unlearning of explicitly provided, known PI attributes_. Each sample is a QA pair where the answer contains the **exact ground-truth PI**, enabling models to explicitly memorize and subsequently attempt to forget this information.
> Thus, author-profiling datasets such as SynthPAI cannot be directly repurposed for unlearning auditing because their structure, purpose, and evaluation paradigms are fundamentally different.
>
> In summary, FPI is, to our knowledge, the **first dataset explicitly constructed for evaluating personal-information unlearning**, featuring:
> - Multiple PI attribute types
> - Precise, attribute-specific evaluation metrics
> - A clean forget–retain partition
> - Synthetic design that ensures controlled, reproducible auditing scenarios
> These capabilities are not provided by TOFU, SynthPAI, or existing PI-inference datasets.

---

> ### Author Response · Authors · 2025-11-28
>
> >- The paper is a bit over-claimed. Considering the auditing method may be limited to PI datasets (or similar tasks with verifiable answers) and fine-tuning unlearning methods, it is not a unified auditing method.
> >-  Can the auditing method be applied to broader tasks (e.g., open-ended generation), or other types unlearning methods?
>
> We thank the reviewer for the comments regarding the scope of our auditing method and the concern about over-claiming. We address each point below.
>
> **Scope of the Proposed Auditing Method**
>
> The reviewer asks whether the proposed auditing method can be applied to broader tasks (e.g., open-ended generation) and suggests that the method might not be unified across unlearning scenarios.
>
> As stated in our **Introduction**, this work explicitly focuses on **personal-information (PI) unlearning**. All contributions are scoped to this setting, and the paper does **not** claim that our method provides a unified auditing framework for all forms of LLM unlearning or all generation tasks. PI unlearning constitutes a well-defined and practically important problem, distinct from knowledge unlearning in both structure and evaluation. We therefore believe that developing PI-specific auditing methodology fills an essential gap rather than representing a limitation.
>
> Extending auditing to broader tasks such as open-ended generation is an important **future direction**. The key challenge in that direction is to develop appropriate metrics for unlearning and recovery. This challenge is so significant that it in its own right a complex research problem. As it is our future goal to conquer that problem, this present work serves as the first proof of concept for that mission.
>
>
> **Misinterpretation About Limitation to Finetuning-Based Unlearning**
>
> The reviewer raises the concern that our auditing method may only apply to finetuning-based unlearning. We clarify that our focus on finetuning-based unlearning is **not** due to any limitation of the proposed auditing framework. Rather, in a white-box setting, auditing prompt-based and decoding-based unlearning is **straight-forward**, since these methods leave the model parameters unchanged and the original information can be immediately recovered, as discussed in Section 3 (line 162).
>
> More formally, consider the notation in our paper:
> - Prompt-based unlearning produces a model $f^u :=(h^u, \pi^o, g^o)$ where an input modification strategies (e.g., adding a system prompt instructing the model not to answer) is applied, while the underlying transformer $\pi^o$ and the decoding strategy $g^o$ is unchanged.
> - Decoding-based unlearning yields a model $f^u :=(h^o, \pi^o, g^u)$, where only the decoding function is altered, but still the base model $\pi^o$ is unchanged.
>
> In a white-box auditing setting, the auditor is permitted to modify any components of $f^u$. Thus,
> - For prompt-based unlearning, replacing the modified input function with an identity map $h^{\rm id}(x)=x$ (i.e., removing the injected prompt) yields $f'=(h^{\rm id}, \pi^o, g^o)$, instantly revealing the supposedly removed information.
> - For decoding-based unlearning, replacing $g^u$ with a standard decoding strategy (e.g., standard greedy decoding) gives $f'=(h^{o}, \pi^o, g^{\rm greedy})$, again fully recovering the original information.
>
> Because $\pi^o$ is unchanged in both cases, the information is **never removed—only masked**. Therefore, these approaches do **not** satisfy the GDPR “right to erasure,” and auditing them is a straight-forward exercise, possessing little research value.
>
> **Why finetuning-based unlearning requires auditing**
>
> In contrast, finetuning-based unlearning modifies the model parameters: $f^u :=(h^o, \pi^u, g^o)$,
> with $\pi^u \ne \pi^o$. In this case, it is challenging to determine whether the information has been erased or is still implicitly encoded in $\pi^u$. This is precisely where white-box auditing—and therefore our proposed method—is significant.
>
>
>
>  **Summary**
> - Our contributions are claimed within the clearly stated scope of **PI unlearning**.
> - The method is not inherently limited to finetuning-based unlearning; we focus on it because prompt- and decoding-based approaches easily fail white-box auditing.
> - Broader tasks such as open-ended generation are outside the intended scope but represent meaningful directions for future work.

---

> ### Author Response · Authors · 2025-11-28
>
> >-  The proposed method is lack of empirical score to comprehensively quantify the ability for a given unlearning method. Besides, the idea itself lacks stronger (even theoretical guarantee), and it is intuitive for rigorous auditing task.
> >- The proposed auditing method needs stronger guarantee.
>
>
> The reviewer raises concerns about (i) the lack of a comprehensive empirical score for auditing, and (ii) the absence of theoretical guarantees. We address both points below.
>
> **Empirical quantification of unlearning effectiveness**
>
> In the revised manuscript, we have added a **rigorous formal definition** of white-box auditing (**see Section 5 Line 281-302**), where we define a notion of $\alpha$- effectiveness and $\beta$-robustness quantify the auditing outcome for a given unlearned model by explicitly measuring both the unlearned model's original forget quality and its forget quality when a recovery method is applied.  We hope this addresses the concern.
>
> **Theoretical guarantee**
>
> Deriving theoretical guarantees in the context of **LLMs** is extremely challenging. The high-dimensional, nonlinear, and opaque nature of transformer architectures makes it intractable to obtain formal guarantees without imposing unrealistic assumptions. This is precisely why, to date, even membership-inference–based auditing has only been studied under simplified statistical settings.
>
> Given these limitations, we argue that a realistic and robust auditing protocol for LLM unlearning may consider adopting a **toolbox perspective**, where multiple complementary recovery strategies are designed for different types of unlearning. Such auditing can operate in a **red-team / blue-team loop**:
> - red team: develops tailored recovery strategies to expose remaining leakage;
> - blue team: improves unlearning to close those gaps.
> Our work provides **one such recovery tool** (i.e., the RIG method), serving as a proof of concept that existing finetuning-based unlearning, even when applied on a synthetic data, can leave recoverable information. Expanding this toolbox is an important direction for future research.

---

> ### Author Response · Authors · 2025-11-28
>
> >-  The key finding that existing unlearning methods can not robustly erase the target knowledge is not a new conclusion, many recent works have reveal this point. However, the authors fail to fully discuss and compare with them.
>
> We thank the reviewer for pointing this out. In the original manuscript (Related Work, Line 111 onward), we already discuss prior findings showing that supposedly forgotten knowledge may remain recoverable. Below, we provide a more detailed clarification of how our setting fundamentally differs and why our contribution remains novel.
>
> **Prior findings were established under different settings**
>
> Although recent works have shown that forgotten information can often be recovered, these results were obtained under **knowledge-unlearning** settings. The underlying assumptions, data format, and unlearning objectives differ substantially from **personal-information (PI) unlearning**, which is the focus of our work.
>
> For example, _Zhang et al. (2025)_ demonstrated that **quantization** of an unlearned model’s parameters may inadvertently restore erased knowledge. This quantization-based recovery satisfies our white-box auditing requirements. However, their experiments were conducted on **knowledge-unlearning datasets**, where factual knowledge is entangled and can re-emerge even after coarse parameter perturbations.
>
> In contrast, PI unlearning has fundamentally different characteristics --- PI attributes are _heterogeneous, uncorrelated_, structured QA pairs have _unique ground truth_, and forget/retain partition has _no entanglement_. It is therefore **not obvious** that quantization-based recovery would succeed for PI tasks.
>
> Indeed, in our experiments (Table 2), we evaluate quantization-based auditing on the FPI dataset and find that it provides **only limited recovery ability**. Without our study, one might incorrectly conjecture that PI unlearning is similarly vulnerable based on previous knowledge-unlearning results, or conversely, falsely conclude that PI unlearning is already effective based purely on the quantization-based auditing results. Our results show that neither conclusion holds.
>
>
> **Prior “relearning-based” recovery differs from white-box auditing**
>
> Other relevant works (Lynch et al., 2024; Hu et al., 2025; Deeb & Roger, 2025) analyze unlearned models under a **relearning setting**, where a set of auxiliary data that either correlated with or drawn from the forget set is provided. The unlearned model is then "relearned" on these samples. They show that forgotten knowledge can resurface through relearning. However, this setting is not comparable with ours, since our white-box auditing setting assumes **no such auxiliary forget-related data** is available.
>
> **Novelty of our contribution**
>
> To our knowledge, **no prior work has examined whether PI information—across heterogeneous attributes—remains recoverable under white-box auditing**. Our evaluation demonstrates that PI unlearning exhibits different behaviors from knowledge unlearning, and that existing methods fail in ways previously undocumented.
>
> We will incorporate this expanded discussion into the final version to improve clarity.

---

> ### Author Response · Authors · 2025-11-28
>
> >- What are new findings compared to previous works, e.g., Inexact Unlearning Needs More Careful Evaluations to Avoid a False Sense of Privacy?
>
>
>  We thank the reviewer for bringing up this paper.
>
> The paper _Inexact Unlearning Needs More Careful Evaluations to Avoid a False Sense of Privacy_ studies unlearning from the perspective of **membership inference**: given a sample, the adversary aims to determine whether it was included in the unlearning process.
>
> This problem setup is fundamentally different from the one studied in this work.
>
> **Membership inference vs. information reconstruction**
>
> Membership inference asks: _Was this specific sample used for training or unlearning?_
> This evaluates whether an attacker can distinguish between:
> - a truly unseen sample, and
> - a sample that was originally seen and later unlearned.
>
> In contrast, our work targets a **much more challenging** notion of auditing:
> _Can the content of the forgotten personal information be reconstructed from the model after unlearning?_
>
> This is a substantially more demanding criterion. Even if a model defeats membership inference (i.e., the attacker cannot tell whether a sample was used during unlearning), the model may still internally encode and reveal the supposedly erased personal information.
>
> **The novelty of our work with respect to membership inference**
>
> A successful membership inference attack only shows that the model behaves differently on a sample that underwent unlearning.
> It **does not** demonstrate that:
> - the underlying private attribute can be recovered,
> - the information persists in internal representations, or
> - the unlearning mechanism fails to remove (rather than merely hide) the information.
> Our findings are therefore new.
>
>
> To our knowledge, no prior work—including the above paper—has examined whether **structured personal information** can be _reconstructed_ from a model after unlearning under a **white-box auditing setting**. Our work exposes a different and more challenging auditing problem than membership inference.
>
> We will add these clarifications in the final manuscript.

---

### Official Review · Reviewer_gB1T · 2025-10-28

**Soundness:** 2
**Presentation:** 2
**Contribution:** 2
**Rating:** 4
**Confidence:** 4

**Summary:**

This paper studies whether current unlearning methods for large language models could erase sensitive personal information (PI) or merely suppress it. The authors propose a white-box auditing framework and introduce a synthetic Fake Personal Information (FPI) dataset that covers four attribute types (year of birth, blood type, postcode, social insurance number). They design restricted inverse greedy (RIG) decoding, which selects the least likely token from an attribute-specific restricted candidate set, to probe whether information targeted by finetuning-based unlearning remains latent in model logits.

**Strengths:**

1. The research problem is clear and timely. White-box auditing of PI unlearning aligns with real regulatory needs (e.g., GDPR) and provides a rigorous test of actual erasure vs suppression.
2.  FPI spans heterogeneous attribute types (numeric, categorical, structured sequences) and defines tailored metrics, enabling nuanced evaluation.

**Weaknesses:**

1. Experiments use only DeepSeek-7B with LoRA; robustness across model architectures and sizes is not assessed.
2. No comparison to retraining-from-scratch to quantify the ideal unlearning behavior and establish an upper bound on true erasure.
3. Previous unlearning audit methods often employed approaches with statistical guarantees, such as membership inference attacks, whereas the method proposed in this paper appears intuitive.

**Questions:**

- Can the authors replicate on additional open LLMs (e.g., Llama-3, Mistral, Qwen) and different sizes to test generality?
- How does the proposed audit method work on retraining, and does it produce false positives?
- Does the proposed method have theoretical guarantees to ensure the rigor of the audit?

---

> ### Author Response · Authors · 2025-11-28
>
> > - Experiments use only DeepSeek-7B with LoRA; robustness across model architectures and sizes is not assessed.
> > - Can the authors replicate on additional open LLMs (e.g., Llama-3, Mistral, Qwen) and different sizes to test generality?
>
> We thank the reviewer for this valuable suggestion. To strengthen the empirical evaluation and assess robustness across different model families and scales, we have reproduced the results of Figure 2 using an additional open LLM, Qwen3-8B, which differs from DeepSeek-7B both in architecture and size.The new experimental results, consistent with our earlier resutls, are included into the revised manuscript Appendix F Figure 7.
>
>
> > - No comparison to retraining-from-scratch to quantify the ideal unlearning behavior and establish an upper bound on true erasure.
> > - How does the proposed audit method work on retraining, and does it produce false positives?
>
> We thank the reviewer for this insightful comment. In the revised manuscript, we have added new experiments to compare against an ideal retraining-from-scratch baseline, which provides a reference on the behaviour of true erasure.
>
> Specifically, we finetuned a DeepSeek-7B model on the FPI dataset **excluding all forget data**, and treat this model as the gold-standard retrained model, denoted as $f^ *$. Since $f^ *$ has never seen the forget samples, its behavior represents the ideal unlearning target.
>
> For $f^ *$, we report in Figure 2 both (i) its **original forget quality** (red bar) and (ii) its forget quality after applying **RIG** (green bar). When RIG is applied, the forget quality of $f^ *$ remains unchanged for postcode and SIN, shows only a slight decrease for blood type, and slight increases for year of birth. The notable fluctuations are likely due to statistical irregularities.
>
> In contrast, when RIG is applied to models unlearned using GA, GD, and GA+KL, we observe a **consistent and substantial drop** in forget quality across all four attributes. Moreover, the RIG-induced forget quality scores for these unlearned models are **always lower** than both the original and RIG-applied scores of $f^ *$.
>
> Together, these results demonstrate that RIG effectively reveals  that residual personal information remained in the unlearned models. The slight drop of forget quality on $f^ *$ for unlearning blood type (Figure 2a) does not seem to demonstrate the existence of "false positives" in the recovery, given that such drops are not observed in other attributes.
>
> Similar results are also observed for Qwen3-8B model (see Appendix F Figure 7.)
>
> >- Previous unlearning audit methods often employed approaches with statistical guarantees, such as membership inference attacks, whereas the method proposed in this paper appears intuitive.
> >- Does the proposed method have theoretical guarantees to ensure the rigor of the audit?
>
> We appreciate the reviewer’s observation and agree that prior unlearning audits commonly rely on membership-inference–based tests that come with statistical guarantees in that specific setting. We would like to stress that our work targets a _strictly stronger_ notion of privacy: instead of auditing whether a particular example was used during training/unlearning (membership), we assess whether an auditor can _reconstruct the information that was intended to be removed_. This reconstruction-based criterion goes beyond membership inference and is significantly more challenging to characterize theoretically, especially in the context of large language models where model outputs are high-dimensional, generative, and non-deterministic. Techniques designed for membership inference do not directly extend to our setting, and existing theoretical frameworks provide limited traction.
>
> Agreeably providing rigorous theoretical guarantees for reconstruction-based unlearning audits in LLMs is an important future direction. We believe our empirical study offers a first practical step in this direction and our results, capable of recovering forgotten information, further justify the importance of this direction.

---

### Official Review · Reviewer_1DQL · 2025-10-29

**Soundness:** 3
**Presentation:** 3
**Contribution:** 2
**Rating:** 4
**Confidence:** 3

**Summary:**

This paper proposes a white-box auditing framework to verify whether personal information claimed to be forgotten through unlearning methods is genuinely removed. The authors construct a synthetic dataset (FPI) containing fake personal information and construct a Restricted Inverse Greedy (RIG) decoding strategy to recover supposedly forgotten information. The experimental results reveal that current unlearning approaches often fail to fully eliminate sensitive information, with recovery rates reaching up to 97% for some attributes.

**Strengths:**

1) The paper is well-written with clear motivation, methodology, and results.
2) The proposed RIG decoding approach seems simple yet effective.
3) Unlearning auditing is a critical privacy concern in deployed LLMs.

**Weaknesses:**

1) The entire evaluation is conducted on artificially generated fake personal information. Real-world personal data often has different statistical properties, correlations, and contextual dependencies that may affect both unlearning effectiveness and recovery difficulty. The generalizability of findings to real privacy-sensitive scenarios remains unclear.
2) The paper dismisses prompt-based and decoding-based unlearning methods too quickly, focusing exclusively on finetuning-based approaches, which significantly narrows the contribution.
3) The paper provides simple explanations for why RIG works but lacks formal theoretical analysis. Why does gradient ascent necessarily push information into low-probability regions? Under what conditions will RIG succeed or fail? What are the theoretical guarantees or limitations?
4) It is insufficient to make evaluations on a single model.

**Questions:**

1) Can you provide formal theoretical analysis or guarantees for when and why RIG is effective? Specifically, under what loss landscapes or model properties will gradient ascent-based unlearning push information into low-probability regions that RIG can exploit?
2) Can the proposed auditing method work on different types of unlearning methods?
3) Regarding real-world scenarios, can the proposed dataset simulate different statistical settings (such as bias, where a personal information dataset may exhibit a long tail on certain features, and where forgetting high-frequency and low-frequency features presents differing challenges)? Additionally, can the auditing method be robustly applied to real-world considerations?

---

> ### Author Response · Authors · 2025-11-28
>
> >- The entire evaluation is conducted on artificially generated fake personal information. Real-world personal data often has different statistical properties, correlations, and contextual dependencies that may affect both unlearning effectiveness and recovery difficulty. The generalizability of findings to real privacy-sensitive scenarios remains unclear.
>
>
> We thank the reviewer for this comment and acknowledge the gap between synthetic and real-world personal data.
>
> **Why synthetic PI is necessary.**
> Due to strict privacy regulations (e.g., GDPR, CCPA), using or releasing real personal information for unlearning research is generally infeasible. As a consequence, no publicly available real-world PI-unlearning dataset exists, and synthetic data becomes a _necessary requirement_ for reproducible and ethically compliant research.
> Furthermore, no suitable synthetic dataset for PI unlearning currently exists, which motivates the construction of our FPI dataset.
>
> **Challenges in modeling real-world distributions.**
> We agree that real personal data contains complex correlations and contextual dependencies that are difficult to faithfully replicate. The absence of real PI datasets means that there is **no ground-truth reference** from which to derive principled statistical modeling guidelines. Thus, any synthetic dataset will inevitably diverge from real-world distributions to some degree.
>
> Our goal, therefore, is not to perfectly mimic real PI distributions but to provide a controlled and transparent benchmark that enables rigorous measurement of unlearning effectiveness.
>
>
> **Why synthetic results are still meaningful.**
> Even with synthetic data, the core insights of our work remain highly relevant to real-world settings:
> - The _white-box auditing protocol_ we propose can serve as a practical governance mechanism independent of data realism.
> - RIG illustrates a realistic attacker behavior—_leveraging any available prior knowledge_ to reconstruct supposedly forgotten information.
> - The findings reveal a _concrete vulnerability in existing unlearning methods_ , showing that even under simplified conditions, PI can be recovered.
>
> **Toward a more comprehensive auditing toolbox.**
> We envision real-world white-box auditing to operate not through a single universal recovery strategy, but through a **toolbox of task-specific methods**, used in a red-team vs. blue-team fashion. Our work provides one such tool and a corresponding evaluation framework. Extending this toolbox to more complex data distributions is an important direction for future research.

---

> ### Author Response · Authors · 2025-11-28
>
> >- The paper dismisses prompt-based and decoding-based unlearning methods too quickly, focusing exclusively on finetuning-based approaches, which significantly narrows the contribution.
> >- Can the proposed auditing method work on different types of unlearning methods?
>
> We thank the reviewer for raising this point. Below we clarify why prompt-based and decoding-based unlearning are not the focus of our study and why white-box auditing becomes straightforward in these settings.
>
> **Why prompt-based and decoding-based methods are excluded.**
> As discussed in Section 3 (line 162), these two families of methods **do not modify the model parameters** and therefore do not remove the target information from the model. Instead, they prevent the model from _rendering_ the information during inference, effectively **hiding** it rather than unlearning it.
>
> More formally, consider the notation in our paper:
> - Prompt-based unlearning produces a model $f^u :=(h^u, \pi^o, g^o)$ where an input modification strategies (e.g., adding a system prompt instructing the model not to answer) is applied, while the underlying transformer $\pi^o$ and the decoding strategy $g^o$ is unchanged.
> - Decoding-based unlearning yields a model $f^u :=(h^o, \pi^o, g^u)$, where only the decoding function is altered, but still the base model $\pi^o$ is unchanged.
>
> **Why white-box auditing is straightforward for these methods.**
> In a white-box auditing setting, the auditor is permitted to modify any components of $f^u$. Thus,
> - For prompt-based unlearning, replacing the modified input function with an identity map $h^{\rm id}(x)=x$ (i.e., removing the injected prompt) yields $f'=(h^{\rm id}, \pi^o, g^o)$, from which the target information is directly recoverable.
> - For decoding-based unlearning, replacing $g^u$ with a standard decoding strategy (e.g., standard greedy decoding) gives $f'=(h^{o}, \pi^o, g^{\rm greedy})$, again allowing the model to output the original information.
>
> Because $\pi^o$ is unchanged in both cases, the target information essentially remains encoded in the model. These approaches therefore **do not satisfy the GDPR notion of the “right to erasure”**, since the information is not removed from the model, only masked during inference.
>
> **Why finetuning-based unlearning requires auditing.**
> In contrast, finetuning-based unlearning modifies the model parameters: $f^u :=(h^o, \pi^u, g^o)$,
> with $\pi^u \ne \pi^o$. It is _not_ clear whether the target information remains encoded in $\pi^u$, and empirical auditing is necessary to verify unlearning effectiveness.
>
> We will add these discussions in the final revision to improve clarity.

---

> ### Author Response · Authors · 2025-11-28
>
> >- The paper provides simple explanations for why RIG works but lacks formal theoretical analysis. Why does gradient ascent necessarily push information into low-probability regions? Under what conditions will RIG succeed or fail? What are the theoretical guarantees or limitations?
> >- Can you provide formal theoretical analysis or guarantees for when and why RIG is effective? Specifically, under what loss landscapes or model properties will gradient ascent-based unlearning push information into low-probability regions that RIG can exploit?
>
> We thank the reviewer for the insightful questions.
>
> **On the difficulty of providing theoretical guarantees.**
> RIG is evaluated in the context of LLM unlearning, where the model involves highly nonlinear architectures, nonconvex loss landscapes, and complex token interaction patterns. Deriving formal guarantees for when gradient-ascent based unlearning pushes information into low-probability regions—or when recovery is possible—is currently intractable for modern LLMs. This complexity is a fundamental barrier, not a limitation of our specific method.
>
> The primary aim of this paper is to empirically demonstrate a concrete vulnerability in existing unlearning algorithms and to establish a _white-box auditing framework_ within which such vulnerabilities can be systematically evaluated. A complete theoretical characterization of RIG would require strong modeling assumptions that do not hold for LLMs and would detract from the empirical focus of the paper.
>
> Nonetheless, we agree that understanding when RIG succeeds or fails is an important direction and we view this as an important direction for future work. A meaningful path is to conduct formal theoretic analysis with simplified model families (e.g., linear or shallow nonlinear model) to establish conditions under which RIG-style recovery is provably possible.
>
> While such theoretical development is valuable, it requires substantial additional efforts and its complexity makes it a stand-alone research problem in its own right.
>
> >4. It is insufficient to make evaluations on a single model.
>
> We thank the reviewer for this comment. We've reproduced the results of Figure 2 using an additional model, Qwen3-8B. The new experimental results, consistent with our earlier resutls, are included into the revised manuscript Appendix F Figure 7.
>
> >5. Regarding real-world scenarios, can the proposed dataset simulate different statistical settings (such as bias, where a personal information dataset may exhibit a long tail on certain features, and where forgetting high-frequency and low-frequency features presents differing challenges)? Additionally, can the auditing method be robustly applied to real-world considerations?
>
>
> Thank you for this insightful comment. Indeed, features with different frequencies may exhibit significantly different behavior in unlearning and auditing. It will be interesting to fully explore this direction. However, setting up the experiments properly will entail significant effort in the design of forget/retain set partition in combination with varying feature frequencies. Given the limited time for rebuttal, it is infeasible to complete this task in time. We will leave this for a future work.

---

### Official Review · Reviewer_KUmZ · 2025-11-01

**Soundness:** 3
**Presentation:** 3
**Contribution:** 2
**Rating:** 4
**Confidence:** 4

**Summary:**

This paper proposes a white-box auditing framework to evaluate whether personal information unlearning in LLMs is genuine or superficial. The authors build a synthetic Fake Personal Information dataset and design a novel Restricted Inverse Greedy (RIG) decoding strategy that explores low-probability logits to recover supposedly erased information. Experiments on DeepSeek-7B with five popular unlearning algorithms (GA, GD, GA+KL, PO, NPO) show that up to 97% of the forgotten content can still be recovered, revealing that most current unlearning methods fail to fully remove sensitive data and instead merely suppress its surface accessibility.

**Strengths:**

1. The paper offers a white-box auditing method that goes beyond surface-level black-box evaluations of unlearning.


2. The proposed RIG/RG decoding provides a simple way of revealing residual memorization hidden in low-probability regions.


3. Experiments on the FPI dataset are systematic, showing evidence that current fine-tuning–based unlearning methods often only suppress, rather than erase, sensitive information.

**Weaknesses:**

1. The proposed RIG auditing method relies heavily on the design of the restricted candidate set (e.g., top-k or probability threshold), making its results highly sensitive to hyperparameter choices. Small variations in these values can cause large differences in recovery performance, raising concerns about robustness and reproducibility.


2. The approach is mainly effective for low-entropy or format-constrained information (such as numbers, postcodes, or IDs). It is unclear how well it would generalize to unlearning of open-ended or semantically rich knowledge, where answers are expressed in natural language rather than within a fixed token pattern.


3. The evaluation framework is built entirely on a synthetic dataset with clearly defined personal information fields, which simplifies the task but limits realism. The lack of experiments on more diverse or real-world data makes it difficult to assess how the method performs under complex or noisy distributions.


4. The method is diagnostic rather than principled. It does not provide a formal definition or guarantee of what constitutes successful unlearning, nor does it explore theoretical foundations for why RIG should correspond to genuine memory recovery. As a result, while the findings are empirically interesting, their broader implications remain uncertain.

**Questions:**

See weaknesses.

---

> ### Author Response · Authors · 2025-11-28
>
> >1. The proposed RIG auditing method relies heavily on the design of the restricted candidate set (e.g., top-k or probability threshold), making its results highly sensitive to hyperparameter choices. Small variations in these values can cause large differences in recovery performance, raising concerns about robustness and reproducibility.
>
> We thank the reviewer for this observation. We clarify that the design of the restricted candidate set is **intentional and aligned with real-world threat modeling**, rather than a methodological weakness.
>
> **Use of prior knowledge is inherent to realistic PI-recovery attacks.**
> The restricted candidate set is designed to model attackers who leverage **any available prior knowledge** of a target attribute to strengthen recovery attempts. This mirrors realistic privacy threats, where adversaries do not rely on a single generic method but instead tailor their strategies. Therefore, sensitivity to candidate-set construction is _expected_ and highlights an important point:  _existing unlearning algorithms can be vulnerable when adversaries use even modest domain knowledge._
>
> In this sense, the candidate-set mechanism is not a drawback but a feature that exposes realistic attack surfaces.
>
> **Why a single “hyperparameter-free” recovery method is insufficient.**
> We agree with the reviewer that robustness is important. However, a _single_ general-purpose recovery algorithm is unlikely to capture the diversity of PI-unlearning scenarios. Given the varied forms of personal attributes (numeric, categorical, structured) and the differing behaviors of unlearning methods, an effective auditing framework should consist of **multiple specialized recovery strategies**, each exploiting different aspects of the model or data.
>
> Our current RIG method serves as **one component** in such a toolbox. It demonstrates that even simple white-box priors can reliably recover supposedly forgotten attributes, highlighting the need for a broader auditing suite.
>
> **Reproducibility and future auditing tools.**
> To ensure reproducibility, our paper provides:
>
> - a fully specified white-box auditing protocol,
> - clear descriptions of candidate-set selection rules, and
> - empirical evidence showing consistent leakage across multiple unlearning methods.
>
> We view RIG as a **proof of concept** demonstrating a concrete vulnerability. A promising future direction is to extend the auditing toolbox by leveraging other white-box signals (e.g., retain-set behaviors, unlearning traces, LLM's latent representations) to construct additional recovery strategies and make the auditing protocol even more comprehensive.
>
> Finally, we note that in the revised version, we have clearly defined a notion of robustness: $\alpha$-effective $\beta$-robustness (see our reply to your comment 4 below and section 5 of our revised paper), and stress that the purpose of white-box auditing in this context is to provide a vulnerability lower bound for an unlearned model.
> In this light, the auditor is allowed to exhaust a wide range of auditing algorithms with diverse hyper-parameter settings so as to recover as much unlearned information as possible. By definition, the auditor is not restricted to a single algorithm or hyper-parameter setting to all unlearned models in different unlearning tasks.
>
> >2. The approach is mainly effective for low-entropy or format-constrained information (such as numbers, postcodes, or IDs). It is unclear how well it would generalize to unlearning of open-ended or semantically rich knowledge, where answers are expressed in natural language rather than within a fixed token pattern.
>
> We appreciate the reviewer’s observation. We agree that RIG is specifically designed for **PI unlearning**, where many attributes are low-entropy or follow constrained formats, and we do not claim that the method generalizes to open-ended or semantically rich knowledge.
>
> However, we emphasize that such generalization is **not the goal of this paper**. Our contributions focus on the PI-unlearning setting, which has distinct characteristics and practical importance. Specifically:
> 1. **Understanding PI unlearning**: We aim to evaluate whether existing unlearning algorithms can reliably remove _personal information_, not generic knowledge.
> 2. **Establishing a PI-unlearning auditing framework and benchmark**: We introduce a principled evaluation protocol with heterogeneous PI attributes and attribute-specific metrics.
> 3. **Providing RIG as a proof-of-concept recovery strategy**: RIG demonstrates that even in this constrained PI setting, current unlearning methods exhibit concrete vulnerabilities. Our goal is not to propose a universal recovery method but to highlight the need for a broader and more systematic study of PI-unlearning auditing.
>
> In summary, RIG is intentionally scoped to PI unlearning, and the paper’s contributions lie in defining, auditing, and analyzing this specific yet critical problem domain.

---

> ### Author Response · Authors · 2025-11-28
>
> >3. The evaluation framework is built entirely on a synthetic dataset with clearly defined personal information fields, which simplifies the task but limits realism. The lack of experiments on more diverse or real-world data makes it difficult to assess how the method performs under complex or noisy distributions.
>
>
> We thank the reviewer for this concern. We agree that real-world PI would add value; however, we emphasize that **using or releasing real personal data for unlearning research is generally infeasible** due to strict privacy regulations (e.g., GDPR, CCPA). As a result, **no publicly available real-world PI-unlearning dataset currently exists**, making a synthetic dataset not merely a design choice but a _necessary condition_ for reproducible research in this domain.
>
> The FPI dataset includes clearly defined and controllable personal-information fields. We view this as a _strength_, not a drawback:
> - It allows **precise ground-truth tracking** of which attributes should be forgotten.
> - It enables **clean, attribute-specific evaluation metrics**, which are essential for rigorous unlearning auditing.
> - It avoids ethical risks associated with handling real human data.
>
>
> Regarding realism, we agree that real-world PI is more complex and noisy; however, **there is no publicly available reference dataset** to guide such complexity modeling. In this context, we believe that constructing synthetic data with controllable structure is a more scientific approach than attempting to approximate the features of unavailable real PI by arbitrary speculations.
>
>
> **FPI contains diversity.**
> Although synthetic, FPI includes meaningful diversity:
> - multiple heterogeneous PI types (numeric, categorical, structured sequences),
> - intentional personal information entanglement (e.g., groups of individuals sharing last names), which introduces nontrivial correlations that unlearning methods must handle.
>
> This diversity is sufficient to demonstrate the core insight of the paper: **current unlearning algorithms can leak personal information even under controlled white-box auditing.**
>
> We agree that expanding FPI to incorporate additional complexity is a promising direction, and we highlight this as part of future work. Nonetheless, we believe the current design provides adequate variety for evaluating PI-unlearning vulnerabilities and establishing the importance of white-box auditing.

---

> ### Author Response · Authors · 2025-11-28
>
> >4. The method is diagnostic rather than principled. It does not provide a formal definition or guarantee of what constitutes successful unlearning, nor does it explore theoretical foundations for why RIG should correspond to genuine memory recovery. As a result, while the findings are empirically interesting, their broader implications remain uncertain.
>
> We thank the reviewer for this insightful comment.
>
> **Formalization of unlearning auditing.**
> In the revised manuscript, we have added a **rigorous formal definition** of white-box auditing (**see Section 5 Line 281-302**), where we define a notion of robustness to assess the vulnerability of a unlearned model.
>
>
> To that end,  let $\ell(\mathbf{y},\mathbf{y}') \in [0,1]$  be a loss function measuring the error between a candidate text sequence and a reference text sequence. For any dataset $D:=\{ (\boldsymbol{x} ^ {(i)}, \boldsymbol{y} ^ {(i)}) \} _ {i=1}^N$, the average loss of a model $f$ is
> $R(f, D):= \frac{1}{|D|}\sum\limits_{(\boldsymbol{x}, \boldsymbol{y})\in D}\ell(f(\boldsymbol{x}), \boldsymbol{y})$
>
> Let $U$ be an unlearning algorithm, which takes  $(D_{\rm rtn}, D_{\rm fgt}, f^o)$ as input and generate an unlearned model $f^{u} = U(f^{o}, D_{\rm fgt}, D_{\rm rtn})$. We allow $D_{\rm rtn}=\emptyset$ (e.g., the gradient ascent algorithm). Let $D_{\rm fgt}^{\cal X}$ denote a version of $D_{\rm fgt}$ containing only prompts, with responses removed. We define  an **white-box auditing** algorithm ${\cal A}$ as a function that takes as input $(f^{u}, U, D_{\rm rtn}, D_{\rm fgt}^{\cal X})$ and outputs a model $f^{r}={\cal A}(f^{u}, U, D_{\rm rtn}, D_{\rm fgt}^{\cal X})$.
>
> We say the unlearned model $f^{u}$ is **$\alpha$-effective $\beta$-robust** if $R(f^{ u}, D_{\rm fgt})\ge \alpha$ and if there exists a white-box auditing algorithm ${\cal A}$ such that $R(f^{r}, D_{\rm fgt}) \le \beta$. Note that $\alpha$-effective $\beta$-robustness indicates a lower bound of the unlearned models' vulnerability against white-box auditing. For the same effectiveness value $\alpha$, an $\alpha$-effective $\beta$-robust model with a lower $\beta$ value exposes to a higher risk of information leakage.
>
>
>
> **On theoretical guarantees for RIG.**
> Providing theoretical guarantees for RIG is challenging, due to the highly complex and nonlinear nature of LLMs.  Nevertheless, we view this as an important direction for future work. A promising avenue is to study **toy model families**—where assumptions about the loss landscape, model architecture, and unlearning mechanisms can be made explicit—to establish conditions under which RIG-style recovery is provably possible.
>
> Such analysis, while valuable, requires substantial additional effort and its complexity makes it a stand-alone research problem in its own right.

---

### Meta-Review · Area_Chair_gEHj · 2026-01-06

**Summary:**

While the authors did a commendable job closely reading the reviewers suggestions and improving their manuscript wherever possible, some fundamental issues raised by the reviewer's remains:
- Generalizability beyond structured candidate set design (KUmZ and oink)
- Lack of extensive analysis of false positives (gB1T)

**Reviewer Concerns:**

- Reviewer KUmZ
 - RIG method relies heavily on restricted candidate set design and hyperparameters - response argues this sensitivity is a feature simulating attackers with prior knowledge and RIG is one tool in a broader toolbox.
 - Approach limited to low-entropy data and may not generalize to open-ended knowledge - response clarifies the scope is specifically Personal Information (PI) unlearning which is inherently structured and distinct from generic knowledge.
 - Evaluation on synthetic dataset limits realism and ignores real-world distributions - response states using real private data is legally infeasible and synthetic data is necessary for reproducible ground-truth evaluation.
 - Method is diagnostic rather than principled without theoretical guarantees - response added formal definitions of auditing robustness to the revision but notes theoretical guarantees are intractable for complex LLMs.

- Reviewer 1DQL
 - Synthetic data lacks statistical properties and correlations of real personal data - response reiterates real PI is unavailable due to privacy laws but synthetic data effectively demonstrates current methods leak information.
 - Prompt-based and decoding-based unlearning methods dismissed too quickly - response explains these methods only mask info without modifying parameters so white-box auditing can trivially recover it.
 - Lack of theoretical analysis for why gradient ascent pushes info into RIG-accessible regions - response acknowledges this is interesting but deriving guarantees for deep non-linear LLMs is beyond the scope of this empirical study.
 - Insufficient evaluation relying on a single model - response added experiments using a second model Qwen3-8B which produced consistent results.

- Reviewer gB1T
 - Evaluation limited to one model architecture and size - response reproduced results on Qwen3-8B to demonstrate robustness across different architectures.
 - No comparison to retraining-from-scratch to establish upper bound for erasure - response added a gold-standard baseline trained from scratch and showed RIG extracts significantly less info from it.
 - Method lacks statistical guarantees found in membership inference attacks - response argues information reconstruction is a stricter privacy standard than membership inference and theoretical guarantees are currently infeasible.

- Reviewer oink
 - Dataset lacks novelty and is similar to TOFU or SynthPAI - response distinguishes FPI as designed for unlearning known structured attributes unlike TOFU's entangled knowledge or SynthPAI's author profiling.
 - Paper overclaims contribution as it is not a unified auditing framework - response clarifies scope is explicitly limited to PI unlearning and fine-tuning methods where auditing is non-trivial.
 - Method lacks empirical scores to quantify ability and needs stronger guarantees - response points to new formal definitions of robustness added to the revision to quantify auditing outcomes.
 - Finding that unlearning is not robust is not new and lacks comparison - response notes prior works focused on generic knowledge or membership inference while this uniquely demonstrates structured PI reconstruction.

**Reviewer Scores:**

- Reviewer KUmZ main concern about the auditing strategy's generalizability beyond highly structured PI was not addressed and would have retained **4**
- Reviewer 1DQL concerns were largely addressed (they were perhaps some misunderstandings of the setting being considered) and would have like increased to **6**
- Reviewer gB1T core concern about lack of analysis around TPR and FPR rates was not addressed and would have likely retained **4**
- While some of Reviewer oink's concerns were addressed, their main concern (same as KUmZ) and so would have likely had a score of **4**

This averages to **4.5**

---

### Decision · Program_Chairs · 2026-01-26

Reject